# Bloodstream Infections in Critical Care Units in England, April 2017 to March 2023: Results from the First Six Years of a National Surveillance Programme

**DOI:** 10.3390/microorganisms13010183

**Published:** 2025-01-16

**Authors:** Olivia D. Conroy, Andrea Mazzella, Hannah Choi, Jocelyn Elmes, Matt Wilson, Dimple Y. Chudasama, Sarah M. Gerver, Miroslava Mihalkova, Andrew Rhodes, A. Peter R. Wilson, Nicholas Brown, Jasmin Islam, Russell Hope

**Affiliations:** 1City of Wolverhampton Council, Wolverhampton WV1 1SH, UK; 2UK Health Security Agency, London E14 4PU, UK; andrea.mazzella@ukhsa.gov.uk (A.M.);; 3Tower Hamlets London Borough Council, London E1 1BJ, UK; 4St. George’s Hospital, London SW17 0QT, UK; 5Department of Microbiology, University College London Hospitals, London NW1 2BU, UK; 6Cambridge University Hospitals NHS Foundation Trust, Cambridge CB2 0QQ, UK

**Keywords:** bacteremia, bloodstream infection, antibiotic resistance, critical care, central venous catheters, case-fatality rate, public health surveillance, England

## Abstract

Background: Patients in critical care units (CCUs) are at an increased risk of bloodstream infections (BSIs), which can be associated with central vascular catheters (CVCs). This study describes BSIs, CVC-BSIs, organism distribution, percentage of antimicrobial resistant (AMR) organisms, and case fatality rates (CFRs) over the first six years of a voluntary national CCU surveillance programme in England. Methods: Surveillance data on BSIs, CVCs, and bed-days between 04/2017 and 03/2023 for adult CCUs were linked to mortality and AMR data, and crude rates were calculated. Results: The rates of CCU-BSIs and CCU-CVC-BSIs were stable for the first three years (3.6 and 1.7 per 1000 bed-days in 2019/20), before increasing by 75% and 94% in 2020/21, respectively, and returning to near pre-pandemic levels by 2022/23. Gram-negative bacteria accounted for 50.3% of all CCU-BSIs, followed by Gram-positive bacteria (39.6%) and *Candida* spp. (8.6%). *Klebsiella* spp. saw increases in percentage AMR, whereas other organisms saw declines or similar levels. The overall CFR was 30.2%. Conclusions: BSI incidence in CCUs remained stable across the study period, except for an increase in 2020/21 which reverted by 2022/23. These data provide a benchmark for CCUs and give insight into long-term AMR patterns where comparable national data are limited.

## 1. Introduction

Patients in critical care are at a higher risk of developing healthcare-associated infections due to increased disease severity, comorbidities, the need for invasive procedures, and greater use of antibiotics [1,2,3]. A national point prevalence survey in 2011 and 2016 found that the prevalence of healthcare-associated infections was highest in English adult CCUs (23.1% and 21.2%, respectively) [4].

Central vascular catheters (CVCs) are used in healthcare settings for monitoring and treatment purposes and are particularly common in CCU settings. CVCs are a known risk factor of bacterial and fungal bloodstream infections (BSIs) [5], and BSIs linked to central vascular catheter use (CVC-BSIs) are associated with an increased cost of care and more severe outcomes [6,7].

From 2009 to 2011, a large intervention study known as ‘Matching Michigan’ was implemented in English CCUs. The study aimed to replicate technical and behavioural interventions used in a United States study which saw large reductions in CVC-BSIs [5,8]. A 60% reduction in English CVC-BSI rates in adult CCUs was observed following interventions and system-wide changes, although there was substantial variability in detection, diagnosis, and reporting.

In response, the Infection in Critical Care Quality Improvement Programme (ICCQIP) was established in 2011 as a multidisciplinary collaboration to create a standardised national infection surveillance system for English CCUs. The programme emphasised clinical stewardship to reduce CVC-BSIs. After an initial pilot programme in May 2016, the programme was extended to all English CCUs in November 2016. Data published for the first year identified substantial interunit variation in CVC-BSI rates across the participating CCUs, differences in the primary pathogens isolated compared to those identified in positive blood cultures (PBCs) from the general hospital population, and increased antimicrobial resistance (AMR) [1]. This current study expands on data reported from the first year of surveillance and presents data for the first six full financial years from the national voluntary surveillance programme. This paper describes participation in the programme, changes in BSIs and CVC-BSIs in CCUs over time, causative organisms, AMR profiles, and case fatality rates.

## 2. Materials and Methods

### 2.1. Data Sources

All English CCUs were invited to participate in the surveillance programme; however, participation was non-mandatory. This paper presents data from adult CCUs.

Participating units submitted data on positive blood cultures (PBCs) and aggregate denominators (bed-days data) to an online surveillance platform, the ICCQIP Data Capture System. A unit was deemed as having participated for a given financial year if it had submitted either PBC or bed-days data during a given month, for any 6 months of the year. All other analyses used the full dataset.

PBC case data include patient demographics, date of admission to the CCU, date and time the specimen was taken, organisms identified, repeat blood culture results, CVC information, clinical symptoms associated with the PBC, and likely source of infection. To minimise bias, CCUs submitting data were not asked to determine if a PBC meets a specific case definition (see Table 1). Instead, data were collected to enable categorization of the reported PBC.

Denominator data include total number of occupied patient bed-days, occupied patient bed-days limited to patients in the unit for at least two nights, CVC bed-days limited to patients in the unit for at least two nights, and the number of blood culture sets taken in the unit. The programme began in May 2016, but data were analysed starting from April 2017 to cover the first full financial year of national implementation. Data were extracted on 28 May 2024. More details on the methods of programme data are available in the ICCQIP surveillance protocol [9].

AMR for relevant bacteria and mortality data were acquired by linking to the UK Health Security Agency’s (UKHSA) Second Generation Surveillance System AMR dataset and the UK’s National Health Service (NHS) Spine dataset (a repository for NHS patient information including patient mortality) using NHS numbers and date of birth. Across financial years, between 92.3 and 94.3% of cases were successfully linked to the mortality dataset, and between 64.7 and 78.2% of cases were linked to the AMR dataset.

### 2.2. Analysis

CCU numerator and denominator data are validated each quarter on submission to the ICCQIP Data Capture System. Missing or invalid denominator data were imputed using the nearest previous month’s data from the same unit. For patients with multiple PBCs caused by the same organism reported in the same 7-day window, only the first PBC was retained. Crude incidence rates of each metric (e.g., BSI) were calculated as the number of cases of that metric divided by the relevant denominator (e.g., number of occupied patient bed-days) in each financial year (April to March). Units missing all denominator data for a particular rate calculation were excluded from the analyses, to avoid overestimation of the unit type counts and rates.

Case fatality rates (CFR) were calculated as the number of deaths from any cause reported within 30 days of specimen date as a percentage of all reported cases. For cases where multiple records shared the same NHS number and date of birth within the 30-day fatality window, only the record with the specimen date nearest to the date of death was included in the analysis. Data were processed and analysed using StataSE version 17 (StataCorp LLC, College Station, TX, USA) and R version 4.3.2 (R Foundation for Statistical Computing, Vienna, Austria).

### 2.3. Ethical Approval

All data were collected within statutory approvals granted to the UKHSA for infectious disease surveillance and control. Information was held securely and in accordance with the Data Protection Act 2018 and Caldicott guidelines.

## 3. Results

### 3.1. Participation in the Surveillance Programme

Of the 243 adult CCUs in England, between 30% and 38% each year submitted at least 6 months of data to the programme (Table 2). During 2020/21, units were offered the option of pausing data entry due to the COVID-19 pandemic, and participation declined during this period. Participation improved in the years following the pandemic, returning to pre-COVID-19 levels (from 72 units in 2019/20 to 83 units in 2022/23).

### 3.2. Counts and Rates of Infections

Between April 2017 and March 2023, 282,595 blood culture sets in adult CCUs were reported, of which 21,026 (7.4%) were positive. Among these, 11,548 (54.9%) were classified as BSIs.

Skin commensals were isolated from 47.1% of the total PBCs across the six years (9895/21,026), with coagulase-negative staphylococci (CoNS) making up 40.6% of isolates (9671/23,772). However, only 3.9% of skin commensal PBCs met the BSI definition (387/9895). For a skin commensal PBC to be defined as a BSI, a repeat positive blood sample of the same skin commensal within 48 h of the first specimen is required; this information was only reported in 4.8% of cases (478/9895).

Of the 11,548 BSIs, 7755 (67.2%) were CCU-BSIs (median age 58 years, interquartile range 47–69 years; 65.5% in male patients). Until 2019/20, the rates of CCU-BSIs varied between 3.6 and 4.0 per 1000 bed-days > 2 nights, before peaking at 6.3 in 2020/21 and steadily declining to pre-pandemic levels of 3.9 per 1000 bed-days > 2 nights in 2022/23 (Figure 1). The 75% increase in rates between 2019/20 and 2020/21 was due to a combination of a 62% increase in the number of reported CCU-BSI cases and a 7% reduction in reported bed-days > 2 nights (Table 2). To explore this unexpected decline in reported unit activity and its impact on the estimated rate, we conducted a secondary complete-case analysis of the units that consistently submitted data in every financial year between 2017/18 and 2021/22. Among these units, there was an 86% increase in the number of CCU-BSIs and a 18% increase in the reported bed-days > 2 nights, leading to a 58% increase in rate of CCU-BSIs.

The most common, known sources of CCU-BSIs were pulmonary (2606, 33.6%), CVC (2438, 31.4%), digestive/hepatobiliary (1030, 13.3%), genitourinary (403, 5.2%), and skin or soft tissue (253, 3.3%).

CCU-CVC-BSIs followed a similar temporal trend to CCU-BSIs, fluctuating between 1.7 and 2.0 per 1000 CVC-days > 2 nights between 2017/18 and 2019/20, before rising to a peak of 3.3 in 2020/21 and decreasing steadily to 2.2 per 1000 CVC-days > 2 nights in 2022/23 (Table 2).

The use of CVCs during CCU stays (CVC utilisation) remained stable between 2017/18 and 2019/20, ranging from 59.1% to 61.7%, before rising to 65.6% in 2020/21. Use of CVCs has since declined to 58.9% in 2022/23 (Table 2).

The blood culture positivity was at an average of 6.5% between 2017/18 and 2019/20, and then it increased to 8.8% in 2020/21 and has since declined to 8.0% in 2022/23 (Table 2).

### 3.3. Reported Organisms

Of the total PBCs, 10.9% were identified as polymicrobial infections in 2017/18, increasing to a peak of 16.7% during 2020/21 and subsequently reducing to 14.1% by 2022/23 (Table 2).

For CCU-BSIs, Gram-negative bacteria accounted for 50.3% of all CCU-BSI isolates, followed by *Enterococcus* spp. (17.8%), *S. aureus* (10.8%), and *Candida* spp. (8.6%). Percentages by organisms remained relatively stable between 2017/18 and 2022/23 (Figure 2); however, during 2020/21, the percentages of *Klebsiella* and *Enterococcus* spp. cases increased sharply, while the percentage of *Candida* spp. and *Escherichia coli* cases slightly decreased. These figures returned to previous trends in 2021/22 (Figure 2).

### 3.4. Antimicrobial Resistance

Over the whole study period, 13% of the *E. coli* PBC isolates with susceptibility results were resistant to piperacillin/tazobactam and gentamicin, while less than 1% were resistant to meropenem. For *K. pneumoniae*., 22% were resistant to piperacillin/tazobactam, 8% against gentamicin, and 2% against meropenem. On the other hand, 15% of the *P. aeruginosa* isolates were resistant to meropenem, while 7% were gentamicin resistant (Appendix A). Of the *E. faecium* isolates, 22% were resistant to vancomycin, while only 1% of coagulase-negative staphylococci were. Of the *S. aureus* isolates, 8% were resistant to meticillin or equivalent (Appendix A).

The resistance profiles of some bacteria changed over the years (Figure 3 and Appendix A). The percentage of resistant *K. pneumoniae* isolates increased for to piperacillin/tazobactam (14% to 26%) and amoxicillin/clavulanate (30% to 34%), and for *E. cloacae* to piperacillin/tazobactam (17% to 49%) and ceftazidime (34% to 46%). The percentage of resistant *E. coli* isolates to amoxicillin/clavulanate decreased (56% to 46%), and also for *P. aeruginosa* to ciprofloxacin (16% to 6%).

### 3.5. All Cause 30-Day Mortality

Of the 7141 CCU-BSIs with mortality outcomes, 2160 died within 30 days of diagnosis, giving an overall CFR of 30.2%. Annual CFRs showed a gradual year-on-year increase, from 26.3% in 2017/18 to 33.3% in 2020/21, before decreasing to 26.8% in 2022/23 (Figure 4).

## 4. Discussion

The UKHSA’s national surveillance programme on infections associated with critical care, known as ICCQIP, was established in 2016 in response to the Matching Michigan study to systematically collect data associated with CCU-BSIs in England. The rates of CCU-BSIs and CCU-CVC-BSIs were relatively stable from 2017/18, until both metrics rose to a peak during the COVID-19 pandemic, reaching their highest levels in 2020/21. Following this peak, the CCU-BSI and CCU-CVC-BSI rates declined to near pre-pandemic levels (3.9 and 2.2 per 1000 bed-days > 2 nights in 2022/23, respectively). The overall distribution of causative organisms remained stable across the study period, although there were increases in AMR seen in most organisms between 2017/18 and 2022/23. Mortality rates remained relatively stable.

The rates of CCU-BSIs peaked during 2020/21, with 72 CCUs submitting data for at least six months despite the programme being paused. During this period, the CCU capacity in England increased to accommodate the demands placed by the pandemic [10], but this information was not consistently captured in our data, and therefore CCU denominators were likely underestimated for this period. This may partly explain the 75% rise in the rate of CCU-BSIs reported during COVID-19. However, our secondary complete-case analysis suggested that this rise was not solely attributable to underreported unit activity. Units that consistently submitted data and reported an increase in unit activity still showed a rise in the rate of CCU-BSI, albeit less pronounced.

This trend aligns with the findings of other studies. Research from Rome reported a nearly 3.8-fold increase in the risk of bacterial and fungal BSIs in CCUs during the pandemic compared to pre-COVID-19 levels, attributing key drivers to immune dysregulation in severe COVID-19 cases and lapses in infection prevention and control measures [11]. Immunosuppressive medications may also have factored in the higher risks of acquiring BSIs. A Swedish study reported increased blood culture contamination [12], and data from Paris reported increased risks of acquiring a CCU-BSI in COVID-19 patients compared to those without [13]. Additional factors, such as shifts in patient demographics, increased use of CVCs, under-resourced healthcare systems, and prioritisation of COVID-19 protocols, may have contributed to the rise in CCU-BSIs during the COVID-19 pandemic.

CVC-BSIs are a common occurrence and pose serious complications to healthcare settings globally. In the United States, approximately 150,000 CVC-BSIs occur annually, with approximately 80,000 (53.3%) reported in CCUs [14]. A German point prevalence study estimated that 55% of hospital-acquired BSIs were associated with CVCs [15]. Compared to the CVC-BSI rate reported by Gerver et al. (2020) for the first year of surveillance in the programme (2.3 per 1000 CVC-days > 2 nights) [1], this study reports slightly lower rates for most years apart from 2020/21 and 2021/22. This difference is likely due to having a larger dataset and potential selection bias in the first year of data collection.

A CVC-BSI rate of 1.5 per 1000 CVC-patient days was reported at the end of a 2-year intervention study in English adult CCUs [8], which is lower than that reported here. However, these figures are not directly comparable as the study uses clinical assessments, as opposed to post-collection algorithms, to determine whether a BSI was CVC related [8]. The rise in CVC-BSIs that we was during the COVID-19 pandemic was reported in various countries [16,17,18,19], as well as in the United States, where the CDC reported a significant increase in the incidence of CVC-BSIs during periods of high COVID-19 hospitalisations [20].

Gram-negative bacteria accounted for half of all CCU-BSI isolates, followed by *Enterococcus* spp. (17.8%), *S. aureus* (10.8%), and *Candida* spp. (8.6%). Comparable national data on causative organisms for CCU-BSI data are limited, although data from a 2004 study in the US [21] found that CoNS accounted for the largest proportion CCU-BSIs (35.9%, *n* = 10,515), which is higher than what we report for CCU-BSIs and similar to the percentage we report for all PBCs [21]. Unlike the ICCQIP, the 2004 study did not require a repeat blood culture for confirmation of a BSI but relied on the presence of a CVC, treatment, and other symptoms [21].

Compared to the 2004 study, we report a lower *S. aureus* percentage (10.8% vs. 16.8%), a higher percentage of *Enterococcus* species (17.8% vs. 9.8%) and *E. coli* (10.7% vs. 3.7%), and similar *P. aeruginosa* and *Candida* spp. percentages in CCU-BSIs. A relative increase in Klebsiella and *Enterococcus* spp. CCU-BSIs during 2020/21 was found. Data reported from English hospitals during the COVID-19 pandemic [22,23] also reported an increase in hospital-onset *Klebsiella* spp.; however, a rise in hospital-onset *P. aeruginosa* bacteraemia is not present in our CCU data. The change to organism profiles during COVID-19 may be attributed to changes in infection prevention and control measures, alterations in patient care, and shifts in patient demographics.

The increasing trend in *K. pneumoniae* resistance to piperacillin/tazobactam that we reported between 2017/18 and 2022/23 was also highlighted in the 2023 ESPAUR report among the general population of England (from 15% to 20% resistant between 2018/19 and 2022/23) [24]; of note, the increase in our data occurred before 2021/22 and is therefore not explained by the lowering of the EUCAST ‘R’ breakpoint for *Enterobacterales* in 2021. Conversely, we note that the increasing trend in *K. pneumoniae* resistance to amoxicillin/clavulanate in CCUs was not seen in the general population, where the percentage fluctuated around 30% [24]. The increase in *E. cloacae* resistance to piperacillin/tazobactam in CCUs started in 2019/20, but its further increase in 2022/23 may partly reflect the aforementioned breakpoint change. We also observed small decreases in resistance percentages, which align with what has been reported for overall cases in England: *E. coli* resistance to amoxicillin/clavulanate and *P. aeruginosa* resistance to ciprofloxacin [24]. *E. coli* and *P. aeruginosa* continue to show generally higher resistance profiles in CCUs compared to the general population [24,25], whereas *Enterobacter* spp. in CCU continued to see similar gentamicin and ciprofloxacin resistance compared to the general population [26].

The overall CFR amongst CCU patients with a CCU-BSI was 30.2%, with other studies from English and international CCUs publishing similar figures [27,28]. However, comparability is limited due to the differing causative organisms. The observed increase in the CFR from 2017/18 to 2020/21 is notable, peaking at 33.3% before declining in 2022/23. This gradual rise predates the COVID-19 pandemic, suggesting it is not solely attributable to pandemic-related factors; the reasons behind this trend remain unclear.

The programme and board continue to work on improvement in participation and data quality through continued stakeholder engagement programmes.

## 5. Conclusions

This study summarises the key results from English CCUs between April 2017 and March 2023. BSI rates, case-fatality rates, and causative organisms appear to be relatively stable across the six years, except for changes during the COVID-19 pandemic, which have mostly returned to pre-pandemic patterns. Most organisms showed stable or declining antimicrobial resistance, but resistance to specific antibiotics increased in *K. pneumoniae* and *E. cloacae*. These results offer benchmarking data for CCUs, highlight outliers, and support quality improvement interventions to reduce CCU-BSI and AMR rates. Future research into antibiotic usage and causes of death in CCU-BSI patients could enhance national surveillance efforts.

## 6. Limitations

The COVID-19 pandemic impacted the surveillance of CCU infections, as many units paused reporting. This disruption may have led to changes in data completeness and quality; therefore, estimates of metrics during or shortly after 2020/21 should be interpreted with caution. It is possible that the requirement for a repeat blood culture to classify skin commensals as BSIs in this study led to an underestimation of BSIs, particularly those caused by CoNS. Of note are BSIs caused by *Staphylococcus lugdunensis,* which cannot be distinguished from other CoNS in our surveillance system, despite being recognised as associated with clinically significant infections [29]. Nevertheless, it is understood that this species represents only a relatively small proportion (8.9%) of BSIs caused by CoNS [30].

Additionally, linkage to AMR data was not successful in approximately 30% of cases; if the linked cases were not fully representative of the overall group, this could have led to some bias in the resistance estimates. Finally, case fatality rates were based on death from all causes, as the NHS Spine mortality dataset does not include cause of death, and therefore may be an overestimate of the fatality rate attributable to BSI.

## Figures and Tables

**Figure 1 microorganisms-13-00183-f001:**
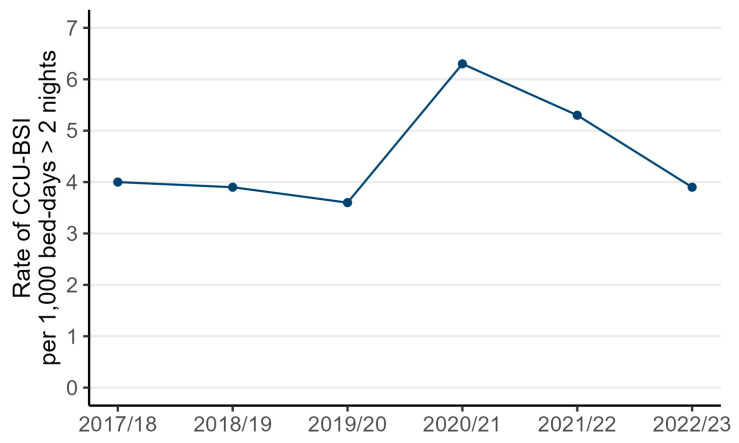
Rates of CCU-BSIs per 1000 bed-days over 2 nights across adult CCUs, England, between 2017/18 and 2022/23.

**Figure 2 microorganisms-13-00183-f002:**
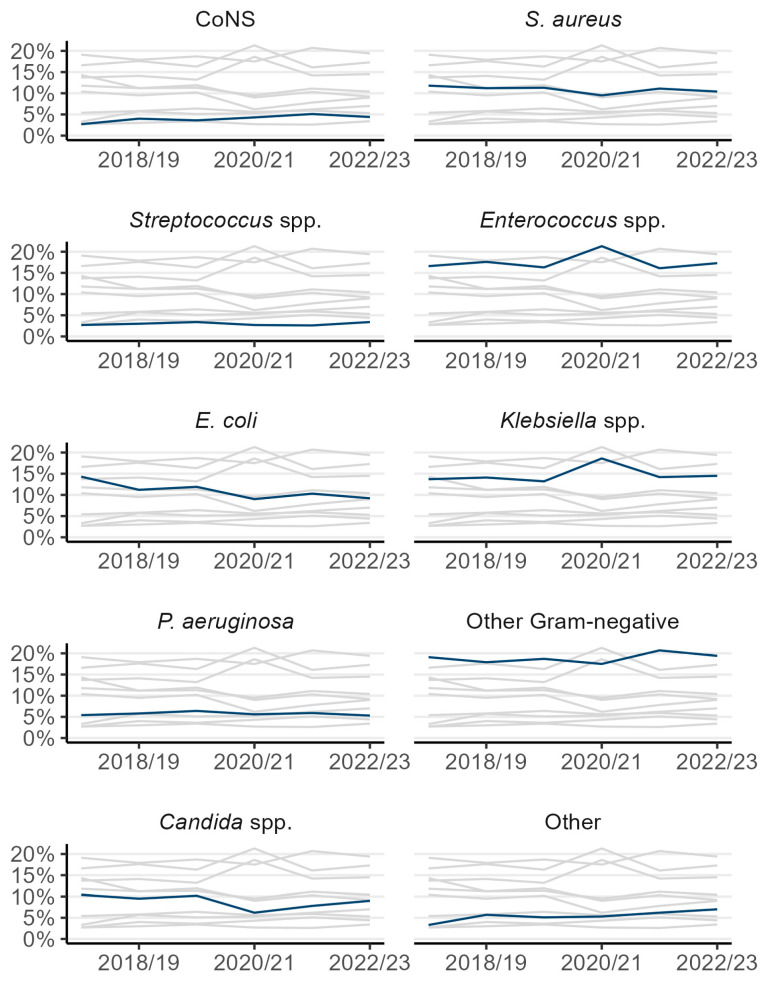
Percentage of organisms identified in CCU-associated BSIs by organism group, adult CCUs, England, between 2017/18 and 2022/23. Each chart shows the same trends, with labelled organism highlighted in dark blue. CoNS: coagulase-negative staphylococci. *S. aureus*: *Staphylococcus aureus*. *E. coli*: *Escherichia coli. P. aeruginosa: Pseudomonas aeruginosa*.

**Figure 3 microorganisms-13-00183-f003:**
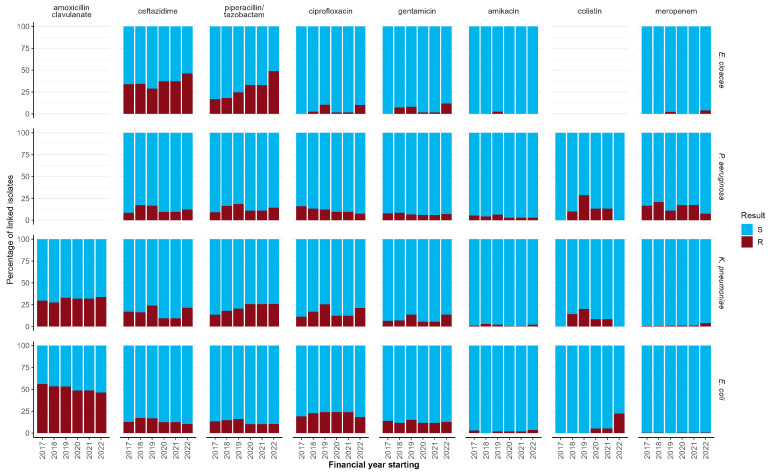
Antimicrobial resistance profiles of *Enterobacter cloacae*, *Pseudomonas aeruginosa*, *Klebsiella pneumoniae,* and *Escherichia coli* PBC isolates to selected antibiotics, by financial year, 2017/18 to 2022/23; participating adult units in England. ‘S’ includes ‘susceptible’, ‘susceptible, normal exposure’, ‘intermediate’, and ‘susceptible, increased exposure’. ‘R’ represents ‘resistant’. Please note that only small numbers of isolates had colistin susceptibility results.

**Figure 4 microorganisms-13-00183-f004:**
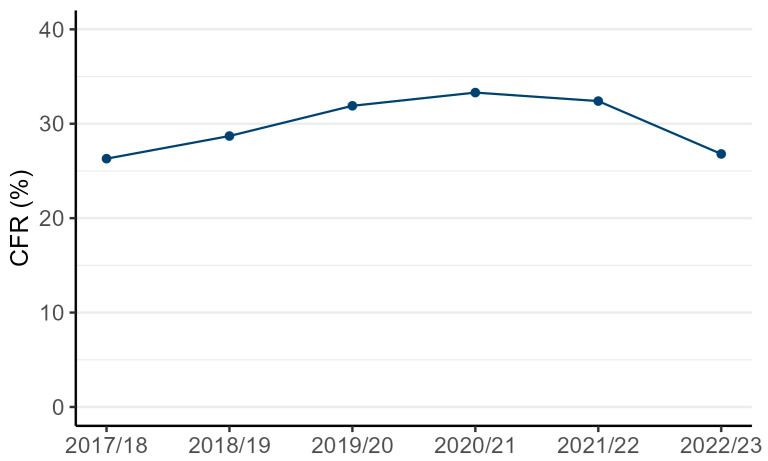
Thirty-day all-cause case fatality rate (CFR, %) of patients in adult units with a CCU-BSI, England, between 2017/18 and 2022/23.

**Table 1 microorganisms-13-00183-t001:** Definitions. * CCU-CVC-BSIs include CCU-BSIs classified as catheter-associated BSIs (CABSIs), catheter-related BSIs (CRBSIs), or both. CABSIs are defined when a CVC is in place at the time of sampling (or removed within 48 h) without evidence of infection at another site. CRBSI occurs under similar conditions but with microbiological evidence of CVC infection or clinical improvement after catheter removal. For further details, see the ICCQIP surveillance protocol [9].

Bloodstream infection (BSI)	- Recognised pathogen identified from at least one blood culture, or - The same skin commensal identified from two blood cultures taken within 48 h and the patient has at least one of the following: fever, rigours, or hypotension.
Polymicrobial infection	Multiple organisms isolated from the same blood culture set, or from multiple blood culture sets in the same day.
CCU-associated bloodstream infection (CCU-BSI)	A BSI is CCU-associated if the patient had been on the CCU for more than 2 nights when the PBC sample was taken.
CCU-associated central-vascular-catheter BSI (CCU-CVC-BSI)	A CCU-BSI that is related to, or associated with, a central vascular catheter *.
Skin commensals	*Aerococcus* spp. *Bacillus* spp. other than *B. anthracis* *Corynebacterium* spp. *Micrococcus* spp. *Propionibacterium* spp. Coagulase-negative staphylococci Viridans group streptococci
Recognised pathogens	All organisms except those classified as skin commensals.

**Table 2 microorganisms-13-00183-t002:** Results by financial year, 2017/18 to 2022/23, for adult CCUs in England. (*) Participation is based on units submitting either case or bed-days data for any six months in that year, as a percentage of the 243 adult units in England.

	2017/18	2018/19	2019/20	2020/21	2021/22	2022/23
Units participating * (%)	79 (33)	84 (35)	92 (38)	72 (30)	78 (32)	83 (34)
Occupied bed-days	354,498	395,530	406,248	359,499	379,761	442,657
Total BC sets taken	44,815	49,269	47,107	45,564	45,865	49,975
Total PBCs	2929	3255	3043	4007	3809	3983
Blood culture positivity	6.5	6.6	6.5	8.8	8.3	8.0
Skin commensals	1249	1481	1336	2003	1793	2033
Percentage of PBCs caused by skin commensals	42.6	45.5	43.9	50.0	47.1	51.0
Count of skin commensals which met the BSI case definition	35	62	50	85	85	70
Count of polymicrobial infections	319	360	387	668	551	561
Percentage of PBCs with a polymicrobial infection	10.9	11.1	12.7	16.7	14.5	14.1
Total BSIs	1674	1839	1727	2154	2098	2056
BSI rate per 1000 bed days	4.7	4.6	4.3	6.0	5.5	4.6
Bed-days for patients in CCU > 2 nights	258,451	287,358	297,049	275,845	282,964	331,120
CCU-associated BSI	1042	1125	1064	1727	1501	1296
Rate of CCU-associated BSI per 1000 bed-days	4.0	3.9	3.6	6.3	5.3	3.9
CCU-CVC Days	152,806	177,385	178,866	181,089	177,115	195,082
CCU-associated CVC-BSI	303	316	308	595	492	424
Rate of CCU CVC-BSI per 1000 CCU-CVC days	2.0	1.8	1.7	3.3	2.8	2.2
CVC utilisation %	59.1	61.7	60.2	65.6	62.6	58.9

## Data Availability

The datasets generated and analysed during the current study are not publicly available due to privacy and confidentiality concerns related to the sensitive health information contained within them.

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
