# Peer review of "Bloodstream Infections in Critical Care Units in England, April 2017 to March 2023: Results from the First Six Years of a National Surveillance Programme"

_microorganisms, 2025, doi:10.3390/microorganisms13010183_

Round 1
Reviewer 1 Report
Comments and Suggestions for Authors
Bloodstream infections in critical care units in England, April 2017 to March 2023: results from the first six years of a national surveillance programme
Title: Appropriate and informative.
Abstract: Appropriate and informative.
The abstract in the text has 210 words; Instructions to the Authors for most of indexed journals maximum 250 words.
Keywords: Please add specific MeSH words like; Surveillance Programme, Bloodstream, Case fatality rates, UK.
Introduction: Appropriate and informative.
Objective: A clear specific objective.
Material and methods: Appropriate and informative; but quite long.
The title of the section could be changed to be “Methodology” OR “Subjects and Methods” OR “Patients and Methods”.
More details are needed for the statistical analysis section.
Table 1: Appropriate and informative.
Results:
Text of the Results: Appropriate and informative.
Figure 1: Appropriate and informative.
Tables 2 and 3: Appropriate and informative.
Figure 2: Appropriate and informative.
Figure 3: Appropriate and informative.
Figure number 3 or 4: Its correct number is Figure 4 “Thirty-day all-cause case fatality rate (CFR, %) of patients in adult units with a CCU-BSI, 213 England: between 2017/18-2022/23”.
Discussion: Appropriate and informative.
Conclusion: Conclusions should be more concise, specific, and related to the results.
Limitations: Appropriate and informative.
Funding: Correct.
Conflicts of Interest: Correct.
References:
Most references are old (>50% of the study references were published before 2019).
Comments on the Quality of English LanguageGood.
Author Response
Thank you for highlighting that the manuscript’s English could be improved. I have been through the entire paper, made many small tracked changes to sentence structure to improve the clarity and readability of the text.
Comment 1 : “Keywords: Please add specific MeSH words like; Surveillance Programme, Bloodstream, Case fatality rates, UK.”
Response 1: Thank you for your suggestions. We have edited the keywords so that they are valid MeSH terms: bacteremia; antibiotic resistance; case-fatality rate; bloodstream infection, public health surveillance, England.
Comment 2: The title of the section could be changed to be “Methodology” OR “Subjects and Methods” OR “Patients and Methods”.
Response 2: Thank you for the suggestion. I had used this title as it was in the journal’s instructions. I will check with the editor whether its possible to use a different title.
Comment 3: Appropriate and informative; but quite long. […] More details are needed for the statistical analysis section.
Response 3: Thank you for your review. I have made some of the sentences more concise. However, it’s difficult to remove detail as we had already removed a lot of methodology with reference to the surveillance system protocol where more detailed methods are found “More details on the methods of programme data are available in the ICCQIP surveillance protocol9”. We have kept what we feel is most relevant to understanding the paper.
Regarding the request for additional details on the statistics, I have reviewed each result to ensure we have adequately described how they were produced. We have provided additional detail on the de-duplication algorithm and the calculation of incidence rates.
Comment 4: “Figure number 3 or 4: Its correct number is Figure 4 “Thirty-day all-cause case fatality rate (CFR, %) of patients in adult units with a CCU-BSI, 213 England: between 2017/18-2022/23”
Response 4: Thank you for highlighting this, we have corrected the mistake.
Comment 5: “Conclusions should be more concise, specific, and related to the results.”
Response 5: Thanks for helping make the conclusions more pertinent. I’ve added concise conclusions on case-fatality rates and AMR resistance and slimmed down the final sentences around context and future research.
Comment 6: Most references are old (>50% of the study references were published before 2019).
Response 6: Thank you for reviewing this. Of the references, 12 were published before 2019, and 16 were from 2019 or later. While I agree that incorporating more recent studies would be ideal, these references reflect the outcomes of a comprehensive evidence search. One of our key findings highlighted the limited availability of comparable data on national BSI surveillance, which we have addressed in the results section: “Comparable international data on causative organisms for CCU-BSI data is limited, although data from a 2004 study in the US21 found that CoNS accounted for the largest proportion…”
Reviewer 2 Report
Comments and Suggestions for Authors
Thank you for inviting me to review this manuscript. It is interesting and very well-written. I have some comments that could be of use:
1. Something is wrong in the authorship. An e-mail is in the authorline. Additionally, there is an email missing at the 4th line of affiliations
2. What is UKHSA? It is all over the affiliations
3. Lines 42-44: Do you have any local data of hospital-acquired infections during the COVID-19 pandemic? This would be of interest given that the study period includes about 3 years after the pandemic took place, while, data from PPSs after the COVID-19 pandemic show higher rates of hospital-acquired infections. If the authors have such data, they could add them in the introduction section
4. Table 1: Were there any Staphylococcus lugdunensis in the isolated species? Even though this is coagulase-negative, it should not be considered a contaminant. If no such discrimination was performed, this should be added in the limitations subsection of the manuscript
5. Table 2: Something is wrong with the number at the last line & first row. You probably mean 59.1?
6. Line 192 and supplementary material: meticillin -> you probably mean methicillin
7. The limitations subsection should be presented at the end of the discussion, right before the conclusions section
Author Response
Comment 1: Something is wrong in the authorship. An e-mail is in the authorline. Additionally, there is an email missing at the 4th line of affiliations.
Response 1: Thank you for highlighting this, I will correct this on the system with the editor.
Comment 2: What is UKHSA? It is all over the affiliations
Response 2: Thank you for your review. I have replaced with the full name, UK Health Security Agency.
Comment 3: Lines 42-44: Do you have any local data of hospital-acquired infections during the COVID-19 pandemic? This would be of interest given that the study period includes about 3 years after the pandemic took place, while, data from PPSs after the COVID-19 pandemic show higher rates of hospital-acquired infections. If the authors have such data, they could add them in the introduction section
Response 3: Thank you for this suggestion. It would be ideal to include this information for comparison, however CCU specific data for overall healthcare-associated infections during this period is not currently available.
Comment 4: Table 1: Were there any Staphylococcus lugdunensis in the isolated species? Even though this is coagulase-negative, it should not be considered a contaminant. If no such discrimination was performed, this should be added in the limitations subsection of the manuscript
Response 4: Thank you for raising the important point regarding Staphylococcus lugdunensis and its potential misclassification in our study. In response, we have updated the limitations section to address this issue. The revised text acknowledges that S. lugdunensis cannot be distinguished from other CoNS in our surveillance system, as there is no separate option for clinicians to specify this.
The updated limitations section now reads:
"The COVID-19 pandemic impacted the surveillance of CCU infections, as many units paused reporting. This disruption may have led to changes in data completeness and quality, therefore estimates of metrics during or shortly after 2020/21 should be interpreted with caution. It is possible that the requirement for a repeat blood culture to classify skin commensals as BSIs in this study is leading to an underestimation of BSIs, particularly those caused by CoNS. Of note are BSIs caused by S. lugdunensis, which cannot be distinguished from other CoNS in our surveillance system, despite it being recognised as associated with clinically significant infections. Nevertheless, it is understood that this species represents only a relatively small proportion (8.9%) of BSI caused by CoNS.
Comment 5: Table 2: Something is wrong with the number at the last line & first row. You probably mean 59.1?
Response 5: Thank you for your detailed review. I have corrected this to 59.1.
Comment 6: Line 192 and supplementary material: meticillin -> you probably mean methicillin
Response 6: Thank you for your comment. The WHO’s International Pharmacopoeia renamed methicillin as meticillin in 2005 (https://doi.org/10.1016/j.jhin.2006.01.001). We appreciate that the old name is still commonly used; however, our preference is to leave the spelling in line the International Pharmacopoeia.
Comment 7: The limitations subsection should be presented at the end of the discussion, right before the conclusions section
Response 7: Thank you for highlighting this. I will check with the editor whether I can change the order of the sections. The current order was taken from the journal template.
Reviewer 3 Report
Comments and Suggestions for Authors
Material and methods
1.BSI definition questionable – “Recognised pathogen identified from at least one blood culture, or:
- The same skin commensal identified from two blood cultures taken within 48 hours and the patient has at least one of the following: fever, rigors, or hypotension” - are you sure that is correct? (recoginsed positive blood culture is BSI? – what about colonizations and contaminations)
2.A CCU-BSI that is related to, or associated with, a central vascular catheter – this definition is not very reproducible, can you please describe more detailed?
Results
3. ICU stand for? CCU was used before, this is same? Same for NHS (I know that almost everybody knows about, but this paper should be readable be everyone.
4.Maybe a list of abbreviations would be useful.
5.No Acinetobacter for CCU-BSI?
https://assets.publishing.service.gov.uk/media/6193c033e90e0704439f4113/hpr1821-acinetobacter20.pdf
6.What percentage of the UK CCU-s was participating the study?
Author Response
Comment 1: 1.BSI definition questionable – “Recognised pathogen identified from at least one blood culture, or:
- The same skin commensal identified from two blood cultures taken within 48 hours and the patient has at least one of the following: fever, rigors, or hypotension” - are you sure that is correct? (recoginsed positive blood culture is BSI? – what about colonizations and contaminations)
Response 1:
Thank you for your comment. The BSI definition used in our manuscript follows the criteria outlined in the ICCQIP (Intensive Care and Critical Care Quality Improvement Program) surveillance protocol. The second part of the definition refers to "the same skin commensal identified from two blood cultures taken within 48 hours," accompanied by at least one of the following symptoms: fever, rigors, or hypotension. This component addresses the distinction between colonization and infection. According to the criteria for the surveillance programme, if a skin commensal (as listed in Table 1) appears in two separate blood cultures within 48 hours and accompanied by clinical symptoms indicative of infection, it provides stronger evidence for a true bloodstream infection, rather than just colonization or contamination. As we report in the results, ‘Only 3.9% of skin commensal PBCs met the BSI definition (387/9,895). For a skin commensal PBC to be defined as a BSI, a repeat positive blood sample of the same skin commensal within 48 hours of the first specimen is required; this information was only reported in 4.8% of cases (478/9,895).’ Therefore, we report that most of the skin commensal infections as contaminants and not BSIs. Finally, please note that CDC and ECDC define BSIs similarly (https://www.cdc.gov/nhsn/pdfs/pscmanual/4psc_clabscurrent.pdf and https://www.ecdc.europa.eu/sites/default/files/documents/HAI-Net-ICU-protocol-v2.2_0.pdf, respectively)
Comment 2: 2.A CCU-BSI that is related to, or associated with, a central vascular catheter – this definition is not very reproducible, can you please describe more detailed?
Response 2: Thank you for your comment on ensuring that the manuscript is reproducible. I have included more detail in the footnotes of Table 1 definitions.
Comment 3: ICU stand for? CCU was used before, this is same? Same for NHS (I know that almost everybody knows about, but this paper should be readable be everyone.
Response 3: Thank you for highlighting this error. I have corrected ICU to CCU and defined NHS.
Comment 4: Maybe a list of abbreviations would be useful.
Response 4: Thank you for the suggestion. I agree that abbreviation lists are very useful, however, the journal format doesn’t allow this from what I can see. I will check with the editor. In the manuscript we do include a table of definitions (Table 1) in an attempt to reduce confusion around the many specific abbreviations.
Comment 5: No Acinetobacter for CCU-BSI?
Response 5: Thank you for your question. We collect data on Acinetobacter as an important pathogen in CCU but do not separate it out as a key group for reporting. In our surveillance protocol, we categorize isolates under broader groups, such as 'Other Gram-negative,' to encompass a variety of Gram-negative organisms that are less common or less frequently reported as specific pathogens. This is in keeping with the grouping we use for our routine surveillance reporting based on the frequency of infections we see for each pathogen.
Comment 6: What percentage of the UK CCU-s was participating the study?
Response 6: Thank you for your question. We have now provided this information in the ‘Participation to the surveillance programme’ section, as well as in Table 2. Please note that the study refers to England only, not to the whole United Kingdom.
Round 2
Reviewer 2 Report
Comments and Suggestions for Authors
The manuscript has been improved.
Reviewer 3 Report
Comments and Suggestions for Authors
Thank you for the explanations and modifications. Paper may be accepted in the present form. Congrats!